# Cobalt Recovery from Li-Ion Battery Recycling: A Critical Review

**Amilton Barbosa Botelho Junior** [1,*], **Srecko Stopic** [2], **Bernd Friedrich** [2], **Jorge Alberto Soares Tenório** [1] and **Denise Crocce Romano Espinosa** [1]

1   Department of Chemical Engineering, Polytechnic School, University of Sao Paulo,
    São Paulo 05508-080, Brazil; jtenorio@usp.br (J.A.S.T.); espinosa@usp.br (D.C.R.E.)
2   IME Process Metallurgy and Metal Recycling, RWTH Aachen University, 52056 Aachen, Germany;
    sstopic@ime-aachen.de (S.S.); bfriedrich@ime-aachen.de (B.F.)
*   Correspondence: amilton.junior@usp.br

**Abstract:** The increasing demand for Li-ion batteries for electric vehicles sheds light upon the Co supply chain. The metal is crucial to the cathode of these batteries, and the leading global producer is the D.R. Congo (70%). For this reason, it is considered critical/strategic due to the risk of interruption of supply in the short and medium term. Due to the increasing consumption for the transportation market, the batteries might be considered a secondary source of Co. The outstanding amount of spent batteries makes them to a core of urban mining warranting special attention. Greener technologies for Co recovery are necessary to achieve sustainable development. As a result of these sourcing challenges, this study is devoted to reviewing the techniques for Co recovery, such as acid leaching (inorganic and organic), separation (solvent extraction, ion exchange resins, and precipitation), and emerging technologies—ionic liquids, deep eutectic solvent, supercritical fluids, nanotechnology, and biohydrometallurgy. A dearth of research in emerging technologies for Co recovery from Li-ion batteries is discussed throughout the manuscript within a broader overview. The study is strictly connected to the Sustainability Development Goals (SDG) number 7, 8, 9, and 12.

**Keywords:** urban mining; hydrometallurgy; deep eutectic solvent; supercritical fluid; nanotechnology

## 1. Introduction

### 1.1. Cobalt: Critical and Strategic Metal

Cobalt (Co) is an essential element of modern life. Its importance made it be considered a critical or strategic metal in the most important economies. Since 2008, Co has been considered a critical material owing to its risk of interruption of supply in the short and medium term (production concentrated in a few countries, political and economic instability, the potential for replacement and recycling rate) and environmental risk (regulations and environmental impact) [1]. Since then, the list has been updated, and the criticality of cobalt is increasing [2]. The last report, released in 2021, classified the element as the tenth critical material among 83 raw materials [3].

The USA classifies Co as a critical (or key) material among 35, because of the risk of interruption of supply in the short and medium term, importance for energy generation by clean technologies, availability in natural resources and production regarding the national and economic security [4,5]. Besides, Brazil classifies Co as a strategic material due to its economic importance, growing demand in the coming decades, application in high-tech products, and dependence for importation or availability of large natural reserves [6].

Currently, the leading producer is the Democratic Republic of Congo (R.P. Congo) in the world, where up to 70% Co is produced, followed by Russia (4.3%) and Australia (4%). Also, R.P. Congo concentrates almost 51% of the reserves worldwide, followed by Australia (20%) [7]. The supply of Co in the world is a concern due to the control of a single country for the most significant production and resources. Additionally, production from primary

sources in R.P. Congo faces several social and environmental issues directly and indirectly, in addition to political instability.

The impacts of pollution were quantified in areas where Co is extracted. In humans, the exposure of contaminated areas was more than 6 times higher than in areas without industrial pollution. In the environment, the impact was over 40 times that of non-industrial areas [8–11]. Deforestation and losses of human life are a consequence of social conflicts in the country. Warfare is also intensified owing to the mining activity [12].

Although all issues are related to production, Co is vital to modern life. For example, in the USA, about 43% of Co is consumed for superalloys production, 10% in cemented carbides, 16% in other metallic applications, and 31% in several other chemical applications [7]. Cobalt-dependent key technologies are batteries, fuel cells, motors, robotics, drones, 3D printing, and digital technologies [3]. In the world, 57% of cobalt is consumed for batteries production, 13% for Ni-based alloys, and 8% for tool materials [13]. The most critical technology that depends on Co is for battery production, mainly Li-ion batteries.

*1.2. Li-Ion Batteries*

The 2019 Nobel laureates for chemistry went to researchers John B. Goodenough, M. Stanley Whittingham, and Akira Yoshino because of the development of Li-ion batteries. Based on energy generation by electrochemical process, in discharging procedure, the batteries accumulate Li ions in the anode material flow in direction to the cathode. Thus, electricity is generated by electrons flowing from the anode to the cathode by terminals. As the process is reversible, the charging process follows the opposite process. Since they were commercially available in 1991, the Li-ion batteries have been used for many applications: cellphones, smartphones, electronic devices, and electric vehicles [14–16].

These batteries have been used in electric and hybrid vehicles due to the high energy density, non-memory effect, low self-discharge rate, long lifespan, and lightweight equipment than other types of batteries [15]. As a result, the production of electric vehicles is increasing to decline the global $CO_2$ emission (the main greenhouse gas) and also to achieve the principal international agreements (Kyoto Protocol, Paris Agreement, and UN Sustainable Development Goals) [17].

The Li-ion batteries configuration can be cylindrical, prismatic, and pouch, which depends on the vehicle producer [18], and they are composed of: an external casing (Fe-Ni alloys or metallic Al: 20–26%), cathode (~27%), anode (~17%), Cu and Al foils and current collector (~13%), polymeric separator (microporous polypropylene or polyethylene: 4–10%), electrolyte (10–15%), and binder (usually PVDF: ~4%) [19].

Usually, the anode material is made of graphite, and there are commercial batteries composed of graphene, titanium oxide ($TiO_2$), and lithium titanate ($Li_4Ti_5O_{12}$). In addition, new anode materials have been studied, such as titanium niobium oxide ($TiNb_2O_7$) and lithium alloys (Lil, $Li_{21}Si_5$), and $LiC_6$ [20]. Al and Cu foils are bonded by the binder to the cathode and anode, respectively, and the separator is important to avoid the direct contact between positive and negative electrodes preventing short circuits. The Li-ions are transferred from the anode to the cathode (and reverse) through the electrolyte [15].

The Li-ion battery is classified according to the cathode material. Table 1 shows the chemistry of commercial materials and their respective characteristics. The main cathodes containing Co are the most important in the market because of high specific energy and charge and discharge cycles. Compared to free-Co cathodes, the number of cycles can be 3-fold higher in Co-cathodes (LFP x LCO), but more expensive. For this reason, several studies and commercial materials are decreasing the Co content in the cathode, such as NMC 811 with the same energy-specific and almost the same number of cycles [15,21,22].

The increasing demand for batteries for electric vehicles will directly impact Co supply. It is expected that the Co demand for electric batteries of vehicles in the European Union will rise 5 times until 2030 and 15 times until 2050 [23]. The main electric vehicles market will be concentrated in the BRICS countries (Brazil, Russia, India, China, and South Africa), Europe, and the USA due to the number of vehicles currently in use and gross domestic

product (GDP). Nowadays, Norway is responsible for 39.2% of the market, followed by Iceland (11.7%), Sweden (6.3%), Netherlands and Finland (2.6% each), and China (2.2%). From 2014 to 2018, the Chinese market increased 3711% [15].

The increasing demand to supply the battery market will put pressure on Co extraction from primary sources in R.P. Congo, the leading global producer, and boost social and environmental issues [24]. In order to achieve the global agreements to reduce the greenhouse gas emission in the transport area, it is necessary the search for new Co sources.

**Table 1.** Summary of chemistries and characteristics of Li-ion batteries [18,21,22,25–28].

| Cathode Type | LCO | LFP | LMO | NCA | NMC |
|---|---|---|---|---|---|
| Chemical formula | $LiCoO_2$ | $LiFePO_4$ | $LiMn_2O_4$ or $LiMnO_2$ | $Li(Ni_{0.8}Co_{0.15}Al_{0.05})O_2$ | $LiNi_{0.33}Co_{0.33}Mn_{0.33}O_2$–NMC111 $LiNi_{0.5}Co_{0.3}Mn_{0.2}O_2$–NMC532 $LiNi_{0.6}Co_{0.2}Mn_{0.2}O_2$–NMC622 $LiNi_{0.8}Co_{0.1}Mn_{0.1}O_2$–NMC811 |
| Structure | Layered | Olivine | Spinel | Layered | Layered |
| Year introduced | 1991 | 1996 | 1996 | 1999 | 2008 |
| Safety | Moderate | Excellent | Very good | Good | Good |
| Energy density | Very good | Good | Good | Excellent | Excellent |
| Power density | Good | Very good | Very good | Very good | Good |
| Lifespan | Good | Very good | Very good | Very good | Very good |
| Cycle lifespan | Good | Very good | Good | Very good | Very good |
| Performance | Very good | Very good | Good | Very good | Very good |
| Cost | Poor | Very good | Very good | Good | Good |
| Market share | Obsolete | Electric bikes, buses, and large vehicles | Small | Steady | Growing (from NMC 111 > NMC 532 > NMC 622 > NMC 811 to no-cobalt chemistries) |
| Specific density (Wh/g) | 200 | 120 | 140 | 245 | 200 |
| Cycles (charge-discharge) | 1000 | 300 | 820 | 950 | 850 |

*1.3. Urban Mining and Challenges for Sustainable Development*

The use of urban waste as a secondary source of materials has been discussed worldwide and already applied for e-waste (waste electronic equipment, or WEEE). Industrial processes for recycling printed circuit boards operate worldwide, mainly for metals recoveries, such as Cu, Ag, Au, Co, Ge, In, and platinum group [29]. The recovery of materials from urban waste is called *urban mining*, which can be defined as "*the set of processes and activities related to the production of secondary raw materials from urban solid waste*".

There are a few processes designed and/or in operation for battery recycling, such as Retriev, Sumitomo-Sony, Recupyl, Akkuser, and Umicore [15,30]. These processes and the development of new routes are essential not only to recycling waste but also to recovering necessary materials and putting them again into the productive sector (Circular Economy). The circular economy strategies are essential for global Co supply by Li-ion batteries. Despite replacing LCO for NMC batteries (lowest than 5% of Co content), the NMC and NCA type batteries market is expected to dominate from 60% by 2030 to 100% by 2050 [31]. For this reason, allied to the increasing production of electric vehicles boosted by international policies, Li-ion batteries will be critical secondary sources of Co. These patents do not provide information on the degree of purity of the cobalt at the end of the process.

Current recycling processes are driven by two routes: pyrometallurgical (thermal), hydrometallurgical (aqueous), or pyro + hydrometallurgical. For instance, the Umicore process aforementioned recovered Co following the steps: first, the batteries are heated in

a shaft furnace from 300 °C to 1400 °C, where Al-Li slag is separated from the Fe-Co cast; then, two-leaching steps are necessary for $CoCl_2$ recovery [30].

Despite the high capacity of battery recycling to obtain Co in comparison to other processes (7000 tons/year), pyrometallurgical processes have been replaced by hydrometallurgical owing to (i) lower energy consumption; (ii) recovery high-pure products; (iii) possibility to obtain different co-products; and (iv) achieves more goals for sustainable development [15,30,32–37].

The hydrometallurgical route comprises leaching (acid or alkali), purification and separation steps, and product recovery. The process designed by Aalto University for Li-ion batteries recycling is an example of the hydrometallurgy process. After the shredding and sieving of the batteries, the Al foils are removed by alkali leaching. Then, acid leaching under reducing media ($H_2SO_4$ + $H_2O_2$) is used to dissolve the cathode. Purification and separation steps are necessary to obtain a high-pure Co oxalate. Other products are be obtained, such as Al oxide, Al-Cu alloy, Mn oxide, Li and Ni products, and C (anode material) [30].

The current concern about industrial processes being greener and with low environmental impact resulted in leaching and separation techniques for different sources, mainly in urban mining. The innovations are important towards sustainable development and would be combined with the mature technologies and not as a substitute.

The launch of the 2030 Agenda for Sustainable Development Goals (SDGs) shed light upon the future industrial processes and scientific development of the recycling routes, improving the efficiency on the scientific ground [3,23,38,39].

This work aims at the literature review of Co from Li-ion batteries by hydrometallurgical processing. The state-of-art of consolidated techniques are first presented, including acid leaching (inorganic and organic), separation of Co by ion-exchange techniques (solvent extraction and ion exchange resins), and precipitation. Further, the emerging technologies for Co separation are presented (ionic liquids, deep eutectic fluid, supercritical fluids, nanotechnology, and biohydrometallurgy).

## 2. State-of-Art

### 2.1. Leaching of Li-Ion Batteries

Before the selective separation of Co from Li-ion battery recycling, the leaching reaction is essential in the hydrometallurgical route to dissolve the solid phase into the solution. Acids are used to leach Co in the recycling process to dissolve the cathode oxide [40]. There are different reagents reported in the literature, such as inorganic acids and organic acids. The choice of the leaching agent goes beyond process efficiency since the media may impact the separation process. A few examples are further presented.

#### 2.1.1. Inorganic Acid Leaching

The primary consolidated hydrometallurgical technique is the acid leaching by inorganic acids, such as sulfuric acid ($H_2SO_4$), hydrochloric acid (HCl), phosphoric acid ($H_3PO_4$), and nitric acid ($HNO_3$). Equations (1) and (2) show examples of leaching by $H_2SO_4$ without reducing agent, where Co sulfates are formed. Equation (1) depicts the acid leaching of LCO batteries, where Li sulfates are formed, and oxygen gas is released. The NMC 622 type cathode leaching is shown in Equation (2), an example of an NMC battery. The difference among the NMC 111, 811, and 532 in acid leaching is the amount of metallic sulfates generated and acid consumption.

$$4LiCoO_{2(s)} + 6H_2SO_{4(aq)} \rightarrow 4CoSO_{4(aq)} + 2Li_2SO_{4(aq)} + 6H_2O_{(l)} + O_{2(g)} \tag{1}$$

$$20LiNi_{0.6}Mn_{0.2}Co_{0.2}O_{2(s)} + 30H_2SO_{4(aq)}$$
$$\rightarrow 12NiSO_{4(aq)} + 4CoSO_{4(aq)} + 4MnSO_{4(aq)} + 10Li_2SO_{4(aq)} + 30H_2O_{(l)} + 5O_{2(g)} \tag{2}$$

The literature reports the use of reducing agents to increase the kinetic leaching of Co due to the conversion of Co(III) into Co(II), as depicted in Equation (3) (LCO battery) and Equation (4) (NMC battery) by $H_2O_2$ as reducing agent [28,41,42]. Different reducing agents were explored in the literature for chemical reducing, such as sodium bisulfite ($NaHSO_3$) [43], sodium metabisulfite ($Na_2S_2O_5$) [44], and sodium dithionite ($Na_2S_2O_4$) [45,46]. Nevertheless, the Pourbaix Diagram demonstrated that Co would be extracted without the need for reducing agents.

$$4LiCoO_{2(s)} + 6H_2SO_{4(aq)} + 2H_2O_2 \rightarrow 4CoSO_{4(aq)} + 2Li_2SO_{4(aq)} + 8H_2O_{(l)} + 2O_{2(g)} \tag{3}$$

$$\begin{aligned}20LiNi_{0.6}Mn_{0.2}Co_{0.2}O_{2(s)} + 30H_2SO_{4(aq)} + 2H_2O_2 \\ \rightarrow 12NiSO_{4(aq)} + 4CoSO_{4(aq)} + 4MnSO_{4(aq)} + 10Li_2SO_{4(aq)} + 32H_2O_{(l)} + 6O_{2(g)}\end{aligned} \tag{4}$$

Table 2 shows examples of Co leaching by inorganic acids from LCO and NMC batteries from Co extraction, where $H_2SO_4$ and $H_3PO_4$ were used as leaching agents and $H_2O_2$ and $NaHSO_3$ as reducing agents. Extraction efficiency achieved up to 90%. Chen et al. studied the Li extraction to concentrate the solid phase in Co, Ni, and Mn from different Li-ion batteries. Li extraction achieved 93% by $H_3PO_4 + H_2O_2$ mixture with less than 20% Co losses, but leaching parameters would raise the Co extraction. The same authors studied the Co extraction from NMC battery by $H_2SO_4 + H_2O_2$ mixture with 98.5% Co extraction. As stated by the authors, the cobalt, nickel, and lithium products have 99.5% of purity, while manganese product was 93.3%.

Several studies explored the effect of ultrasound in leaching to improve the dissolution of the metal oxides [42,47]. He et al. (2022) evaluated the ultrasound for synthesis of new NMC cathode. In the process, losses of Co (30%) was observed in Li dissolution step. As the authors use pyrometallurgical step previously Li dissolution, the results depicted by the authors clearly demonstrated thermal treatment is not indicated to obtain high pure products [47].

Due to the rarefaction of liquid being irradiated, micro-bubbles can be produced. The implosion of these bubbles can generate high pressure up to 1000atm and transitory high temperature up to 5000K.

Pyro+hydrometallurgical recycling process is also reported in the literature for extraction of Co. According to He et al. (2020), the ammonium sulfate calcination followed by water leaching resulted in Co leaching of 98.5% from the mixture of different types of Li-ion batteries [48]. However, the energy consumption would make the process unfeasible in different parts of the world [49].

**Table 2.** A literature review of Li-ion battery leaching by inorganic acids.

| References | Li-Ion Battery Type | Leaching Agents | Conditions | Co Leaching Efficiency |
|---|---|---|---|---|
| Meshram et al. (2015) [43] | NMC | Leaching agent: $H_2SO_4$ Reducing agent: $NaHSO_3$ | S/L ratio = 20 g/L; 1 M $H_2SO_4$ and 0.075 M $NaHSO_3$; 4 h; 95 °C | 91.60% |
| Chen et al. (2018) [50] | NMC | Leaching agent: $H_2SO_4$ Reducing agent: $H_2O_2$ | S/L ratio = 30 g/L; 1 M $H_2SO_4$; 4% *v/v* $H_2O_2$; 90 min; 70 °C | 98.5% Co |
| Xuan et al. (2021) [51] | NMC | Leaching agent: HCl | S/L ratio = 20 g/L; 4 M HCl; 120 min; 82 °C | ~100% Co |
| Vieceli et al. (2021) [52] | NMC | Leaching agent: $H_2SO_4$ | S/L ratio = 1/50; 2.5 M $H_2SO_4$; 60 min; 50 °C. Calcination at 500 °C for 90 min | 90% Co |
| He et al. (2022) [47] | NMC | Assisted by ultrasound Leaching agent: $H_2SO_4$ | S/L ratio = 10 g/L; 1 M $H_2SO_4$; 250 W; 30 min; 90 °C | 30% Co |
| Takahashi et al. (2020) [42] | LCO | Assisted by ultrasound Leaching agent: $H_2SO_4$ Reducing agent: $H_2O_2$ | S/L ratio = 1/5; pH 3; $H_2O_2$ dosage | 99% |
| Chen et al. (2018) [53] | LCO, LMO, LFP, NMC | Leaching agent: $H_3PO_4$ Reducing agent: $H_2O_2$ | S/L = 1/50; 1 M $H_3PO_4$; 4% *v/v* $H_2O_2$; for 10 min; 40 °C | <20% Co (the goal was the Li extraction) |

2.1.2. Organic Acid Leaching

The literature also reports the use of organic acids for Co extraction from Li-ion batteries. Their use may be considered mature, and many authors report the need for studies on the pilot scale as proof of concept [54]. Acid leaching of LCO type cathode with citric acid ($C_6H_8O_7$) without Equation (5) and with $H_2O_2$ effect Equation (6) are further presented under reducing agent ($H_2O_2$), as well as DL-malic acid ($C_4H_6O_5$) Equation (7) and oxalic acid ($C_2H_2O_4$) Equation (8). After dissolution, the metallic ions form organocomplexes. Equation (9) depicts the leaching of NMC 622 type cathode by $C_6H_8O_7$ under reducing media.

$$6LiCoO_{2(s)} + 6C_6H_8O_{7(aq)} \rightarrow 2Co_3(C_6H_5O_7)_{2(aq)} + 2Li_3C_6H_5O_{7(aq)} + 9H_2O + \frac{3}{2}O_{2(g)} \tag{5}$$

$$6LiCoO_{2(s)} + 6C_6H_8O_{7(aq)} + 6H_2O_2 \rightarrow 2Co_3(C_6H_5O_7)_{2(aq)} + 2Li_3C_6H_5O_{7(aq)} + 15H_2O + \frac{9}{2}O_{2(g)} \tag{6}$$

$$4LiCoO_{2(s)} + 3C_4H_6O_{5(aq)} + H_2O_2 \rightarrow 2CoC_4H_4O_{5(aq)} + Li_2C_4H_4O_{5(aq)} + 4H_2O + O_{2(g)} \tag{7}$$

$$4LiCoO_{2(s)} + 7C_2H_2O_{4(aq)} \rightarrow 2Co(C_4H_4O_4)_{2(aq)} + LiC_2HO_{4(aq)} + 4H_2O_{(l)} + 2CO_{2(g)} \tag{8}$$

$$15LiNi_{0.6}Mn_{0.2}Co_{0.2}O_{2(s)} + 90C_6H_8O_{7(aq)} + \tfrac{1}{2}H_2O_2$$
$$\rightarrow 3Ni_3(C_6H_5O_7)_{2(aq)} + Co_3(C_6H_5O_7)_{2(aq)} + Mn_3(C_6H_5O_7)_{2(aq)} + 5Li_3C_6H_5O_{7(aq)} \tag{9}$$
$$+323H_2O + 114O_{2(g)}$$

There are several advantages of organic acids as an environment-friendly leaching agent compared to inorganic acids. These reagents are considered ecofriendly, easy to degrade, and produce less toxic gases during chemical processes [15,41,55]. For instance, Urias et al. (2020) report the use of lactic, butyric, acetic, and propionic as leaching agents focused on the reduction of $H_2SO_4$ volume, where the mixture achieved similar results for Co extraction than inorganic acids (93.4%) [56].

Musariri et al. (2019) and Esmaeili et al. (2020) demonstrated that Co extraction might achieve over 95% by organic acid leaching of NMC batteries [57,58], similar to reported by Chen et al. (2018) using $H_2SO_4$ and $H_2O_2$ [50]. Ultrasound-assisted leaching is also reported to increase the leaching efficiency, but no changes are found in the literature review.

As reported in Table 3, $H_2O_2$ is commonly used as a reducing agent, but different organic compounds may be used as well, like glucose, lactose, and ascorbic acids. Both could be found as by-products of the agro-industrial process or residues in the case of lactose and glucose. Their use would minimize the environmental impacts of recycling [56]. Ascorbic acid behaves as a vinylogous carboxylic acid and a mild reducing agent and consequently may be used as both leaching and reducing agent [59,60]. As reported by Santhosh & Nayaka (2021), the extraction of Co and Li increased as ascorbic acid was used combined with lactic acid (leaching agent) [61].

The current technological development makes possible the design of industrial processes using organic acids as leaching agents. In addition, separation techniques involving ion exchange and precipitation used in leach solution by inorganic acids can also be used in the organic leach solution. The literature also reports that economic analyses indicate that the process is viable and, shortly, this approach will change from pilot to industrial scale [41,54].

<div align="center">**Table 3.** A literature review of Li-ion battery leaching by inorganic acids.</div>

| References | Li-Ion Battery Type | Leaching Agents | Conditions | Co Leaching Efficiency |
|---|---|---|---|---|
| Musariri et al. (2019) [57] | NMC | Leaching agents: citric acid and DL-malic acid Reducing agents: $H_2O_2$ | 1.5 M citric acid, 2% *v/v*, 30 min, 95 °C; 1.0 M DL-malic acid, 2% *v/v*, 30 min, 95 °C. | Citric acid: 95%; DL-malic acid: 98% |
| Esmaeili et al. (2020) [58] | NMC | Assisted by ultrasound: 37 kHz Leaching agent: lemon juice (citric acid = 90 mg/g, malic acid = 0.86 mg/g, and ascorbic acid 1.24 mg/g); Reducing agent: $H_2O_2$ | S/L ratio = 0.98% (*w/v*); 57.8% *v/v* lemon juice; 8.1% *v/v* $H_2O_2$ | 96% |
| Sun et al. (2018) [62] | NMC (111) | Leaching agent: DL-malic Reducing agent: $H_2O_2$ | S/L ratio = 40 g/L; 1.2 M DL-malic; 1.5% *v/v* $H_2O_2$; 30 min 80 °C. | 94.3% |
| Urias et al. (2020) [56] | LCO | Leaching agents: $H_2SO_4$, lactic, butyric, acetic and propionic; Reducing agents: $H_2O_2$ (6%, *v/v*), and glucose (0.09 mol/L) and lactose (0.09 mol/L) | S/L ratio = 20 g/L; 1.25 M $H_2SO_4$ + organic acids (from a fermentation effluent by an anaerobic microbial consortium) 0.75 M; 18.5 g/L; 300 rpm and 0.09 M lactose from MWP; 86 °C | 93.4% |
| Golmohammadzadeh et al. (2017) [63] | LCO | Assisted by ultrasound Leaching agent: acetic acid Reducing agent: $H_2O_2$ | S/L ratio ratio = 30 g/L, citric acid = 2 M, 1.25% *v/v* $H_2O_2$; 2 h; 60 °C | 81% |
| Nayaka et al. (2016a) [60] | LCO | Leaching agent: glycine Reducing agent: ascorbic acid | S/L ratio = 0.2 g $LiCoO_2$/100 mL; 0.5 M glycine; 0.02 M ascorbic acid; 360 min; 80 °C | 95% |
| Nayaka et al. (2016b) [64] | LCO | Leaching agent: tartaric acid Reducing agent: ascorbic acid | S/L ratio = 0.2 g $LiCoO_2$/100 mL; 0.4 M tartaric acid; 0.02 M ascorbic acid; 5 h; 80 °C | >95% |

## *2.2. Consolidated Technologies for Co Separation*

After leaching Li-ion batteries, several elements are dissolved with Co, such as Li, Ni, Mn, and Al. Therefore, purification and separation techniques are necessary to obtain high-pure products. Solvent extraction, ion-exchange resins, and precipitation are widely used in recycling routes to obtain products from solutions after inorganic and organic acid leaching. Examples are further presented and discussed focused on Co recovery.

### 2.2.1. Solvent Extraction

The solvent extraction technique involves the liquid-liquid separation where the target metal is transferred from the aqueous phase to an organic phase. In the equilibrium, the metal (M) distribution between the phases can be represented as shown in Equation (10). Organic extractants are diluted in kerosene or similar organic compounds. The distribution ratio (D) is calculated as the ratio between the metal concentration in the organic and aqueous phase, and the extraction percentage can be calculated by mass balance [65]. The use of modifiers and mixtures of organic extractants are also used to improve the extraction of metals [32,65,66].

$$[M]_{aq} \Leftrightarrow [M]_{org} \tag{10}$$

The technique is widely used in industry for nickel and cobalt separation and recovery from primary and secondary sources [21,67]. For instance, in nickel laterite processing, the bis(2,4,4-trimethylpentyl)phosphonic acid extractants are used in sulfate media (Cyanex 272, LIX 272, and Conquest 290). In contrast, in chloride media, amine-based extractants are applied (tri-isooctyl amine). The phosphonic acids are cationic organic extractants where cations are released to the solution as the metallic ions go to the organic phase [68]. Figure 1 shows the structure of phosphonic acid and amine-based extractants, which is also used for Co recovery in Li-ion battery recycling.

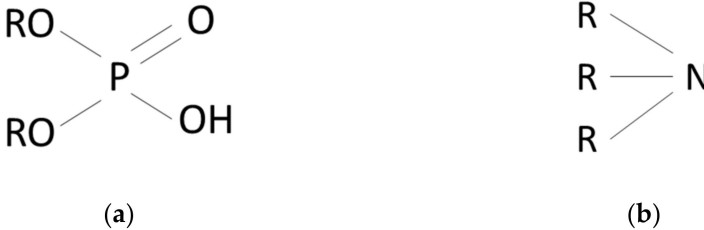

**Figure 1.** Chemical structure of (**a**) phosphonic acid and (**b**) amine-based extractants used in solvent extraction step for Co recovery.

In the process, the leach solution is sent to the stirring tank to contact the organic phase and further separated by density difference. Counter-current extraction setup is commonly used in both academic and industrial proposes. Then, scrubbing is the step for contaminants removal coextracted. Finally, the stripping reaction is the reverse extraction process Equation (10), where the target metal is removed from the organic phase to the aqueous [65,68].

There are several examples in the literature of solvent extraction in Li-ion battery recycling, and the choice of the extractant depends on the leaching agent, as previously explained. For example, Takahashi et al. (2020) studied the Cyanex 272 for Co extraction from the leaching of LCO type cathode by $H_2SO_4$ with higher selectivity than Li. All Li ions remained in the solution, while a high-pure Co solution was obtained [42]. Tsakiridis & Agatzini-Leonardou (2005) studied the effect of Cyanex 272 for separation of Al from Ni and Co in sulfate media (Al foils are used with the cathode as electron collector). According to the authors, Al ions can be removed using Cyanex 272 20% and TBP 5% at pH 3.0 and 40 °C. High-pure Al solution was obtained [69].

According to the manufacturer, there is a selective order of the ions extraction varying the pH of the solution. Among the metallic ions in the bleach solution from Li-ion battery recycling in sulfate media, the extraction of Fe increases from pH 0 to 2, while Al extraction increases from 20% at pH 1 until almost 100% at pH 3. The extraction of Mn and Co is slightly similar, from 20% at pH 2 to 100% at pH 5. The extraction of Ni occurs at pH above 6. In hydrochloric media, the selectivity changes [70]. Theoretically, it is possible to design the process according to this information; however, in a real situation, it will be different owing to the composition, pH, temperature, and aqueous/organic ratio (proportion of leach solution and extractant).

Ichlas & Ibana (2017) evaluated the Cyanex 272 for Co and Ni separation in $HNO_3$ media. The presence of Al and Mn was also evaluated, which compose the NMC batteries. First, Al and part of Co are extracted by Cyanex 272 20% (Extraction I), where the organic phase is scrubbed by sulfuric acid for Co removal from the organic phase. Then, the aqueous phase of extraction I rich in Co and Mn comes into contact with Cyanex 272 20%, where both elements are extracted. Scrubbing and stripping steps are further required to obtain Co and Mn solutions [71]. In the recycling of NMC batteries, the efficiency of solvent extraction reported by Nayl et al. (2015) using Cyanex 272 was 91.2% for Mn, 89.3% for Co, and 95.6% for Ni at pH 3.5, 5.0, and 8.0, respectively. Al removal was carried out using 5-nonylsalicylaldoxime organic extractant (Acorga M5640). The purity of Mn solution was 99.7%, according to the authors [72].

The literature also reports using D2EHPA (bis (2-etylhexyl) phosphoric acid) in the recycling process of Li-ion batteries. Wang et al. (2016) studied the separation of metals after the leaching of cathode material (NMC battery) using $H_2SO_4$ 3 mol/L, $H_2O_2$ 1.6 mg/L, S/L ratio equals to 1/7 at 70 °C for 2.5 h. The Co extraction occurs after the removal of other elements by D2EHPA between pH 2.2 and 2.7. High purity was obtained: 99.5% for Co [73]. Among the elements present in the solution, D2EHPA is more selective for Mn than Co, while the extraction of both elements is associated with using Cyanex 272. As reported by Vieceli et al. (2020) [74], Keller et al. (2021) [75], and Nadimi & Karazmoudeh

(2021) [76], the selectivity of D2EHPA for Mn ions is essential to the process, despite the losses of Co (~10%).

The main challenge of separation by solvent extraction is the costs of organic extractants, industrial equipment, and large areas for high organic/aqueous separation degrees. However, high pure solutions are obtained making the process economically feasible. For this reason, such approach is widely applied, Cyanex 272 and D2EHPA being the most common in both Co extraction from primary and secondary sources.

Figure 2 shows the flowchart of the JX Nippon process for NMC batteries recycling. After the physical process (milling and sieving), the leach solution is obtained after reaction with $H_2SO_4$ at 70–80 °C for 4 h. The solvent extraction separation occurs in three steps. First, Mn is removed using D2EHPA at pH 2.5. Then, Co is separated from the solution at pH 4.2 using PC-88A (2-Ethylhexyl phosphonic acid mono-2-Ethylhexyl ester). Finally, Li-Ni separation occurs at pH 6.5 using PC-88A [15,77,78].

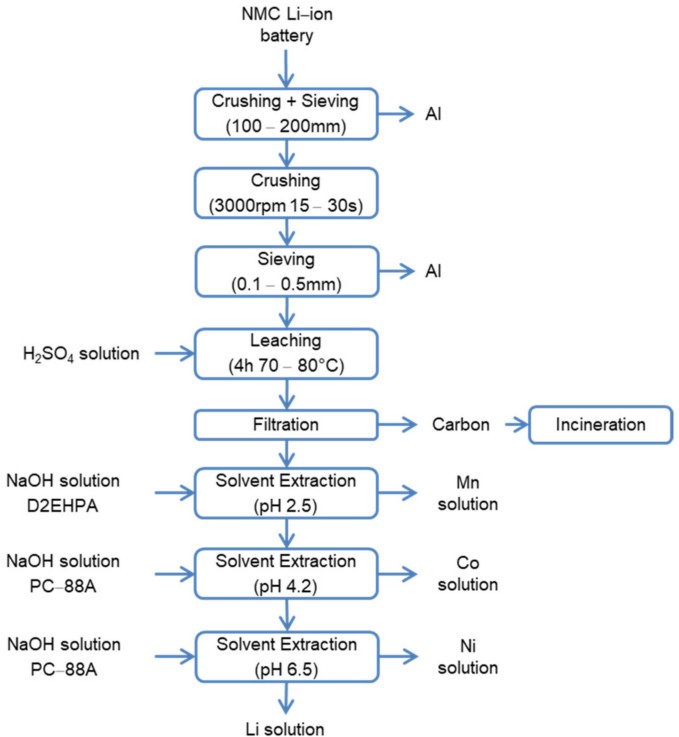

**Figure 2.** Recycling process of JX Nippon Mining & Metals Corporation for NMC Li-ion battery [15,77,78]. Adapted with permission form ref. [15]. 2021 Elsevier.

### 2.2.2. Ion Exchange Resins

Ion exchange resins are another technique reported in the literature for Co separation from the leach solution. In this case, a solid-liquid separation occurs as a reversible exchange between the ions from the aqueous phase (leach solution) and the solid phase (resin). The characteristics of the resins are given by the functional groups, which are cationic (selective for cations) and anionic (selective for anions) resins. Moreover, the chelating resins are used in metallurgical processes due to their order of selectivity. The functional groups react with the metallic ion by coordinate bond interaction or electrostatic interactions [79].

For instance, the theoretical order of selectivity of iminodiacetate functional groups (Lewatit TP 207, Purolite S930, and Amberlite IRC 748, for instance) is as follows: Fe(III) > Cu > Ni > Zn > Co > Fe(II) > Mn > Mg > Al [80–83]. As it occurs in solvent extraction, the separation process can be designed according to the selectivity and composition of the solution [84]. The extraction of Co by iminodiacetate chelating resin increases as

the pH of the solution increase [85,86]. The iminodiacetate functional group is illustrated in Figure 3a.

Also, bis-picolylamine functional groups (Lewatit TP 220 and Dowex M4195, for instance) are reported in the literature for Co recovery, which the theoretical order of selectivity is: Cu > Ni > Fe(III) > Co > Mn > K>Ca > Na > Mg > Al [87,88]. Strauss et al. (2021) report bis-picolylamine chelating resin (Figure 3b) for Ni and Co separation from leach solution of NMC type cathode in the continuous process—Ni is extracted in the first column and Co in the second. The columns connected in series resulted in 99% and 98% of separation efficiencies of Ni and Co, respectively, with low contaminants content [89].

For instance, aminophosphonate acid functional chelating resins (Purolite S950, Amberlite IRC747, and Lewatit TP 260) are also reported as an alternative for separation processes Li-ion battery recycling [90,91]. In this case, the chelating resin is used for Fe, Al, Mn, and Cu removal leaving Co, Ni, and Li in solution [83], for further separation using other chelating resin. The aminophosphonate acid functional group is illustrated in Figure 3c. Considering a process to recycle all types of batteries together (LCO, NMC, NCA, and LFP), the separation steps are critical due to the presence of Fe, which is the primary contaminant in metallurgical processes [88,92–95].

(a)

(b)

(c)

**Figure 3.** Functional groups of chelating resins reported in the literature for Co recovery: (**a**) iminodiacetate; (**b**) bis-pycolilamine; and (**c**) aminophosphonate acid [37,87,96].

In comparison to solvent extraction, the use of resins involves areas smaller and at lower costs. However, the literature reports lower separation degree of target metals (as Co) than impurities. Despite that, several improvements in process parameters have been done to obtain high-pure solutions. Industrial routes involving both solvent extraction and ion exchange resins would be designed to achieve economic feasibility.

### 2.2.3. Precipitation

Precipitation steps may occur in the hydrometallurgical route for separation of the metals or to obtain the final products, or even both [97]. As discussed in solvent extraction and ion exchange techniques, the choice of the reagent depends on the composition of the solution. There are several approaches for precipitation used in hydrometallurgical processes. The use of hydroxides is the most common, which considers alkali reagent to achieve the precipitation value. However, in the case of Li-ion battery recycling, the precipitation of Ni and Co is slightly similar at pH 6.7–8.0, making the selective separation in NMC type cathode batteries difficult by hydroxide [98]. On the other hand, sulfide and carbonate precipitation is commonly reported where this is co-precipitation of Co and Ni [98–100].

As a result, different reagents are used for Co separation in the recycling of Li-ion batteries. In the case of LCO type cathode, where after leaching there is the presence of Co and Li-ions, oxalate acids are used to obtain $CoC_2O_4$ as solid leaving Li ions in solution. The reaction occurs at 50 °C, pH 2.0, and 20% stoichiometric for 1 h [101]. Other authors reports the reaction at 25 °C for 2 h [102]. According to the literature, high pure $CoC_2O_4$ may be obtained.

The literature reports the presence of Mn in LCO type cathode of Li-ion batteries. In this case, Cai et al. (2014) describe the flowchart of the recycling process using precipitation steps. After $H_2SO_4 + H_2O_2$ leaching, Co and Mn were separated from Li using $Na_2S$. The solid was further partially dissolved using acetic acid to obtain CoS. The authors reported high pure salts in all cases [103]. He et al. (2022) depict the oxidative precipitation for Mn removal using $(NH_4)_2S_2O_8$. Then, $CoCO_3$ was obtained using $NH_4HCO_3$, with losses of Co ions in the aqueous phase further recovered as CoS using $(NH_4)_2S$ [104]. The separation of Mn can be carried out also using advanced oxidation processes, such as ozone [105].

Choubey et al. (2021) used $(NH_4)_2S$ for Co-precipitation from NMC type cathode battery in a similar flowchart with co-precipitation of Mn (0.89%) and Li (0.6%). First, after acid leaching by $H_2SO_4$, Ni and Cu were separated by solvent extraction using LIX 84-IC (2 hydroxy 5-nonyl acetophenone). The Co-precipitation occurred at pH 3.0 within 40 min and 10% ($v/v$) of $(NH_4)_2S$ at 25 °C, leaving Mn and Li in the solution [106]. Second, for Co precipitation as oxalate, Chen et al. (2015) separated Ni by precipitation using $C_4H_8N_2O_2$ and Mn by solvent extraction using D2EHPA. The remaining solution contained Li and Co, and $(NH_4)_2C_2O_4$ was used for Co recovery. It was further obtained as carbonate [107].

Several studies have explored the precipitation of all metals present in the leach solution to synthesize new cathode material. For example, Hu et al. (2011) precipitate Ni, Co, and Mn as hydroxide using ammonium solution at pH 11 (NaOH) at 60 °C. Then, LiOH was added in excess. Finally, the solid mixture was heated at 480 °C for 5 h after milling and further calcinated at 950 °C to obtain an NMC 111 type cathode with same purity as the original cathode [108].

Lu et al. (2013) prepared an NMC 811 type cathode by sol-gel methodology from citrate media. The solution containing Ni, Co, and Mn was mixed with ammonium solution at pH 7.0 and heated at 80 °C until the solid phase was formed, which is followed by heating at 450 °C and calcination at 950 °C to obtain the cathode material. The authors also tested the co-precipitation method, which consists of precipitation of all metallic ions using $Na_2CO_3$ and ammonia solution at pH 7.5 and 50 °C for 12 h to obtain $Ni_{0.8}Co_{0.1}Mn_{0.1}CO_3$. The solid was then mixed with $LiNO_3$ and followed the thermal process to synthesize new cathode material [109].

Choi et al. (2021) also used the co-precipitation method to synthesize new NMC cathode material from spent batteries. First, acid leaching was carried out with 2 M of citric acid and 5 wt% glucose assisted by microwave. Then, Ni, Co, and Mn were precipitated as oxalate ($NiMnCoC_2O_4$). Results demonstrated that the pseudocapacitor material has high performance [55]. Thus, there are commercial and environmental advantages for synthesizing cathode material from the spent batteries recycling process since a product with high added value is obtained from a residue.

Figure 4 shows the OnTo Technology flowchart using precipitation steps for the synthesis of cathode material. Duesenfeld GmbH, LithoRec, and Battery Resources "Closed Loop" Process are also examples of recycling process proposing the cathode synthesis from spent Li-ion batteries. The flowchart proposed by Aalto University is a hydrometallurgical route for Co recovery as oxalate [15,30].

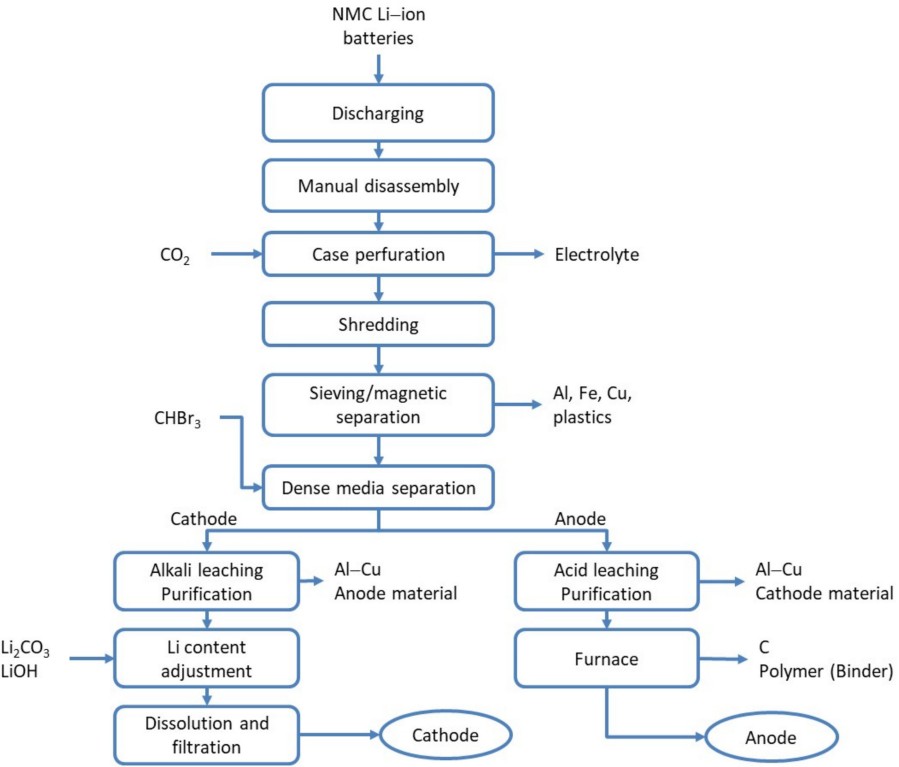

**Figure 4.** Recycling process of OnTo Technology using precipitation steps for synthesis of NMC type cathode from spent Li-ion batteries [15,30]. Adapted with permission form ref. [30]. 2021 Springer Nature.

## 3. Emerging Technologies for Cobalt Separation

As previously reported, the current recycling routes involves acid leaching (inorganic or organic acid) and separation (ion exchange or precipitation). All techniques reported in Chapter 2 are carried out for almost a century. However, the need of industrial process leading with complex residues allied to the sustainable approach puts pressure to the development of new scientific techniques.

The literature reports techniques beyond the state-of-art for Co recovery from the recycling of Li-ion batteries, searching for green techniques towards sustainable development. The use of ionic liquids, deep eutectic fluid, supercritical fluids, nanotechnology, and biohydrometallurgy will be further discussed. Such techniques can be used for leaching or separation or both steps in hydrometallurgical routes. There are a few examples of them strictly related to Li-ion recycling with newsworthy results.

### 3.1. Ionic Liquids

These compounds are liquid salts at room temperature and melting point lower than 100 °C (low flammability) with cations and anions in their chemical structures with a delocalized charge. The ionic liquids also have high thermal stability, negligible vapor pressure, and substantial interaction with organic and inorganic molecules—hydrophilic or hydrophobic, polar or non-polar, and protic or aprotic compounds. Modifying ionic liquids structures is possible by functionalizing with chemical reactions, known as task-ionic liquids [110,111].

A few examples of cations and anions of ionic liquids are presented in Table 4. The combination of these compounds can make the ionic liquid hydrophilic or hydrophobic. For instance, dialkylimidazolium and bis(trifluoromethanesulfonyl)imide is a hydrophobic ionic liquid, while dialkyl imidazolium and tetrafluoroborate is hydrophilic [111,112]. Morizono et al. (2011) studied the ionic liquid alkylhistidine His-EH (histidine-2-ethylhexylamide) diluted

in a mixture of dialkyl imidazolium and hexafluorophosphate (1-hexyl-3-methylimidazolium hexafluorophosphate—[hmim][PF$_6$], and high extraction yield of Co was observed [113].

**Table 4.** A few examples of structures of ionic liquids (R, R1, and R2 = alkyl groups) [110,111].

| Ionic Liquid | Chemical Structure |
| --- | --- |
| Alkylammonium | |
| Phosphonium | |
| Dialkylimidazolium | |
| N-alkyl pyridinium | |
| Trifluoroacetate | |
| Trifluorosulfonate | |
| Tetrafluoroborate | |
| Hexafluorophosphate | |
| Bis(trifluoromethanesulfonyl)imide | |
| Halides: chloride, bromide, iodide | $Cl^-$, $Br^-$, $I^-$ |
| Nitrate, hydrogensulfate, sulfate | $[NO_3]^-$, $[HSO_4]^-$, $[SO_4]^{-2}$ |

The ionic liquids have been widely explored for Co separation in the hydrometallurgical route of Li-ion batteries recycling. Wellens et al. (2012) studied phosphonium ionic liquid (Cyphos IL 101) for Co separation from Ni in hydrochloric media (12 M). A separation factor of 50,000 was observed for Co/Ni using the ionic liquid, while the solvent extraction process using Aliquat 336 was 2500 for the same solution. Stripping of Co was carried out using water [114]. Similar results for Co separation for Ni were found by Parmentier et al. (2016) using Cyphos IL 101 [115]. In LCO recycling, Xu et al. (2020)

demonstrated that the Cyphos IL-101 is more selective for Co than Li ions in hydrochloric media. Co oxalate and Li carbonate have purity of 87.4% and 74.2%, respectively [116].

Zante et al. (2020) studied the separation of metals from the leach solution of NCA type cathode. In this case, imidazolium-based ionic liquids were diluted in tri-n-butyl phosphate (TBP). The hydrophobic ionic liquids 1-butyl-3-methylimi-imidazolium bis(trifluoromethylsulfonyl)imide ($[C_4mim][NTf_2]$) and 1-decyl-3-methylimidazolium bis(trifluoromethylsulfonyl)imide ($[C_{10}mim][NTf_2]$), and sodium salts of tetrakis (trifluoromethyl)phenylboronic (NaTFPB) were studied. According to the authors, Li ions were successfully separated from Ni and Co ions, further separated by different techniques [117]. A stable, supported ionic liquid membrane can be used to separate these ions [118].

Dhiman & Gupta (2019) developed a process for the recycling of NMC type cathode batteries. After leaching by HCl and Mn recovery by precipitation (NaOH), Cyphos IL 102 in toluene was used for Co separation from Li and Ni. The separation efficiency was 99.9% using 0.2 M of ionic liquid without co-extraction. Stripping of Co was carried out using HCl with 100% purity [119].

Zante et al. (2020) evaluated 1-butyl-3-methylimidazolium bis(tri-fluoromethylsulfonyl) imide ($[C_4 mim][NTf_2]$) and tri-hexyl tetradecylphosphonium chloride ($[P_{66614}][Cl]$) ionic liquids and N,N,N',N'-tetra(n-octyl) digly- colamide (TODGA) for separation of metals from leaching of NMC type cathode. First, Mn was separated using TODGA dissolved in $[C_4 mim][NTf_2]$ ionic liquid. Losses of Co was lower than 10% in pH range 0.5–3.75 and O/A ratio lower than 1. The separation of Co ions from Ni and Li was carried out using phosphonic ionic liquid $[P_{66614}][Cl]$ with extraction efficiency of 92.8% without co-extraction [120].

Othman et al. (2020) evaluated the tetraoctylphosphonium oleate $[P_{8888}][oleate]$ for the separation of metals from the leaching of NMC type cathode by HCl. Up to 99% of Co and 89% of Mn were extracted at pH -0.9, where ammonium carbonate removes Mn from the organic phase as $MnCO_3$ leaving Co. The purity of the products achieved close to 100% [121].

Co separation by ionic liquids is pretty much similar or higher than mature technologies, such as solvent extraction and chelating resins. Therefore, these promising reagents can be used in industrial processes due to their safety and environmental appeal. However, many studies are required to understand the separation reaction and costs analysis [112,121,122].

### 3.1.1. Deep Eutectic Solvents

The deep eutectic solvents (DES) are composed of two or more substances that can be self-associated to form a liquid eutectic mixture. DES are classified as a subclass of the ionic liquids with a melting point lower than the substances separated (usually lower than 150 °C), with similar physical and chemical properties to ionic liquids. On the other hand, DES are cheaper, safer, not flammable and volatile, easy to manipulate and synthesize, and considered eco-friendly. For this reason, the literature reports the DES as essential to design a sustainable process [123,124].

Among the DES compounds, the most common are based on choline chloride, carboxylic acids, and other hydrogen-bond donors. Moreover, natural deep eutectic solvents (NADES) include the primary metabolites, namely, amino acids, organic acids, sugars, or choline derivatives [124]. Thus, the mixture of solvents forming the DES consists of hydrogen bond donors and hydrogen bond acceptors. The system is commonly defined as a mixture of Lewis or Brønsted acids and bases with various ionic species. In contrast, ionic liquids have consisted of anion and cations [125].

The literature shows examples of DES for recycling waste electronics and Li-ion batteries in the extraction step [112,126–128]. For example, Tran et al. (2019) reported the recovery of Li and CO from LCO and NMC batteries recycling at 150 °C using DES composed of choline chloride ($C_5H_{14}ClNO$) and ethylene glycol ($C_3H_8O_3$). In addition to

the exciting leaching results, Al foils, binder, and conductive carbon were released without solubilization [129].

Similar results were found by Wang et al. (2019). The authors separated Al foils from NMC type cathode using $C_5H_{14}ClNO$ and $C_3H_8O_3$ at 180 °C. According to Wang et al. (2019), there are more potential applications on an industrial scale for Li-ion battery recycling than ionic liquids—low cost, high efficiency, and low toxicity [130]. At the same conditions and using $C_5H_{14}ClNO$ and $C_3H_8O_3$, Wang et al. (2020) leached up to 95% of Li and Co within 12 h, where $Co_2O_3$ was obtained after reaction between the DES and $H_2C_2O_4$/NaOH [131].

For the direct leaching of NMC type cathode, Co extraction achieves higher efficiency than Ni. At 180 °C, Co extraction achieves over 90%, while Ni reaches 10% during 24 h, and increases over time ($C_5H_{14}ClNO$:$C_3H_8O_3$ equals to 1:2). Extraction of Mn and Li achieve 80% and 70% at 180 °C and 72 h. However, as noted by Schiavi et al. (2021), the reaction with $C_5H_{14}ClNO$ and $C_3H_8O_3$ as DES also leached Cu and Al foils [132], which is not observed in acid leaching by inorganic [42] or organic acids [61]. So, the separation of Al and Cu foils is essential for recycling by DES before cathode leaching.

Roldán-Ruiz et al. (2020) studied the eutectic mixture composed of p-toluenesulfonic acid (PTSA) monohydrate and $C_5H_{14}ClNO$ for Li and Co leaching from LCO type cathode at a temperature lower than 90 °C recovery extraction rates over 94%. In addition, $Co_2O_3$ was obtained after precipitation with carbonate and further calcination [133].

Combinations with organic acids are possible as well. For example, Chen et al. (2021) evaluated the formic acid, and its derived DES (DES-f) prepared to mix $C_5H_{14}ClNO$ and formic acid in molar ratio of 2:1 at 80 °C for 3 h. But, first, Li was extracted using formic acid at 90 °C and 12 h, and DES-f was used for Co leaching at 70 °C [134]. Thus, despite the scarce literature about DES for Li-ion battery recycling and waste electronics, exciting results have justified the development and improvements to design pilot-scale processes.

### 3.1.2. Supercritical Fluids

Supercritical fluids have been tested in recycling processes for extraction due to their specific characteristics: high density, diffusivity, solubility, and reactivity alongside low viscosity make supercritical fluids favorable for extraction and oxidation processes compared to other conventional processes. Carbon dioxide ($CO_2$) has been widely used as a supercritical fluid, for instance, because of its low critical temperature and pressure (31.1 °C and 79.8 bar, respectively) [135].

Similarly, ethane ($C_2H_6$) and ethylene ($C_2H_4$) have their supercritical zone close to the ambient conditions (32.5 °C and 49.1 bar and 9.5 °C and 50.6 bar, respectively), which makes them desirable reagents for industrial and academic purposes. On the other hand, water has a higher critical temperature and similar critical pressure (374.1 °C and 22.1 bar). Supercritical fluids have no surface tension. As in such conditions, there is no interface between gas and liquid. However, the change of pressure and temperature results modify the proprieties of the supercritical fluids for being more similar to gas or liquid. The increase of pressure raises the solubility of the fluid at a constant temperature. At constant density, the solubility increases with temperature. However, it can also decrease with increasing temperature. The critical point is the critical temperature, which will keep the solubility at its lowest [136].

Supercritical fluids have been studied for different applications, such as food science, natural products, pharmaceutical, environmental sciences, and metals recovery. In recycling, the fluids can also be used with solvent extraction technique as complexing agents (with positive or negative charges were) soluble in $CO_2$ supercritical for metallic ions complexation, such as Cyanex 302 (diisooctyl-thiophospinic acid), TBP, and Aliquat 336 [137]. It occurs due to the low solubility of the metals into the $CO_2$ supercritical phase. Also, modifiers such as ethanol and methanol can be used to improve polarity. The extraction of metals by supercritical fluid includes the formation of complex anions with $CO_2$-interface, anion exchange, and mass transfer into the supercritical fluid phase [138,139].

There are a few examples in the literature about recycling Li-ion batteries using supercritical fluids. For example, for Co extraction, Bertuol et al. (2016) compared the acid leaching ($H_2SO_4$ + $H_2O_2$) to acid leaching combined with $CO_2$ supercritical. According to the authors, the leaching of Co achieved 98% in $H_2SO_4$ + $H_2O_2$ system under atmospheric pressure within 60 min, while the extraction using $CO_2$ supercritical reached 95% in 5 min —reduction of reaction time by 12 times and a half of $H_2O_2$ required [140].

The technique can also be used for different approaches than Co extraction. For example, Liu et al. (2016) evaluated the $CO_2$ supercritical for electrolyte recovery from Li-ion battery—ethylene carbonate (EC), dimethyl carbonate (DMC), and ethyl methyl carbonate (EMC). These compounds were solubilized at 15 MPa of pressure and 30 °C within 15 min [141].

Grützke et al. (2014) evaluated the supercritical helium head pressure carbon dioxide (HHPCO$_2$) for separator materials and electrolytes recovery. In this case, the reaction occurred in an autoclave because of the low critical temperature and helium pressure— the experiments occurred at 40 °C and 120 bar. As a result, organic carbonates solvents of LIB electrolytes were recovered and identified (DEC, DMDOHC, EMDOHC, and DE-DOHC) [142].

Fu et al. (2021) reported that the organic binder recovery using $CO_2$ supercritical fluid combined with a cosolvent dimethyl sulfoxide (DMSO) avoided pyrolysis or calcination to release the cathode and Al foils. According to the authors, the recovery efficiency was 98.5% at 70 °C and 80 bar after 13 min of reaction [143]. Thus, the use of superfluid critical has a high potential for Li-ion recycling due to its fast reaction, but several studies are required before scaling up to a pilot scale. Nevertheless, this is one of the most promising emerging technology for recycling processes.

### 3.1.3. Nanotechnology

The last 20 years have shown the countless applications of nanotechnology, which demonstrated that it is the key technology of the 21st century. The recent advances were also observed for extractive metallurgical proposes, mainly for separating the metallic ions. Processes from both primary and secondary sources have been evaluated using nanotechnology, wherein the hydrometallurgical route is called *nanohydrometallurgy*. It has been considered a green technique since the organic extractants are replaced by stable compounds in nanometer-scale ($Fe_2O_3$, ZnO, Ag, and carbon nanotubes, for instance), as previously discussed [144–146].

The nanohydrometallurgy is still lab-scale, and studies are required to achieve the improvements necessary to scale up. Despite that, actual results are found in the literature for the separation and purification of leach solutions. ZnO nanoparticles are commonly used for adsorption of ions due to their low cost, abundant availability, and not-toxic behavior. They can be used for adsorption of some metallic ions the recycling of Li-ion batteries [147].

Le et al. (2019) evaluated the adsorption rates of Cu, Ag, Pb, Cr, Mn, Cd, and Ni under UV effect. According to the authors, for recycling purposes of Li-ion batteries, Cu ions can be separated from Ni and Mn, but under UV light, the adsorption of Mn is higher than Ni. Therefore, it may be applied for the NMC type battery process [148]. In addition, some authors suggested that the ZnO nanoparticles have better adsorption than other metal oxides, such as $Fe_3O_4$ and CuO [147].

The improvement of separation rate by $Fe_3O_4$ nanoparticles can be carried out using protecting coating and complexing agents. For example, Melo et al. (2019) studied the separation of metals of magnetic nanoparticles using $SiO_2$ as protecting coating and the ligand diethylenetriaminepentaacetic acid (DTPA). In this case, the essence of the process is the mixing of the leach solution with the nanoparticles, and, right after the reaction, the solid-liquid separation occurs using a magnet. Then, an acid solution is used to recover the metallic ions from the nanoparticles to the solution. According to the authors, the separation efficiency followed the order: Mn > Co > Ni > Cu > La [149].

Lobato et al. (2019) coated $Fe_3O_4$ magnetic nanoparticles with Cyanex 272 organic extractant for Co separation. The system was also coated with $SiO_2$, and the nanoparticles stayed stable in acid conditions (2 M $H_2SO_4$). According to the authors, the separation efficiency was similar to the solvent extraction process by 25 times faster using magnetic nanoparticles [150].

Also, $Fe_2O_3$ and $Fe_3O_4$ can be coated with oleic acids for Co separation. Compared to Cyanex 272 in traditional solvent extraction separation, nanoparticles' faster kinetic is highlighted [151,152]. Besides, organo-nanoparticles have been explored and attracting attention owing to their high reactivity, chelating functional groups (carboxyl and hydroxyl), chemical stability, and natural raw materials. Yildirim et al. (2020) studied the reaction between Ni and Cu ions with fungus extract-chitosan as bionanosorbents. Selective separation of Cu over Ni and adsorption data similar to the magnetic nanoparticles, chelating resins, and solvent extraction [153].

Maatar & Boufi (2015) studied the poly(methacrylic acid-co-maleic acid) grafted nanofibrillated cellulose (NFC-MAA-MA) aerogel for adsorption of metallic ions, including Ni ions. Efficiency achieved 95%, but studies are necessary to evaluate selective separation of metallic ions [154]. Furthermore, nanomaterials can be used for manufacturing, such as graphene as anode material or coated with cathodes of Li-ion batteries [25,155].

3.1.4. Biohydrometallurgy

The bioprocess, commonly known as biohydrometallurgy, is another way to recycle spent electronic equipment, avoiding landfilling and losing valuable elements. For instance, copper production from primary sources (ores) occurs by bioleaching in Chile [156], and the technology is spread for printed circuit boards [157–160]. As a branch of hydrometallurgy, the process is based on microorganisms in aqueous media for leaching and separation of metallic ions. Several authors considered the technique promising, with potential for industrial-scale and eco-friendly, mainly for low-grade and complex sources, where waste electronics are included [15,37,161–163]

Bioleaching includes three types of processes:

- redoxolysis (reaction occurs as biooxidation and bioreduction)
- acidolysis (proton promotes dissolution with biogenic inorganic or organic acids)
- complexolysis (complexation promoted dissolution).

The leaching efficiency of Li-ion batteries is strictly related to more energy and carbon sources because of the complexity. In separation, the microorganisms (alive or dead) can be used for adsorption (called biosorption). In some cases, the microorganisms can create chelating compounds with metallic ions [164–166].

Bioleaching by *Acidithiobacillus ferrooxidans* is widely used due to iron and sulfur-oxidizing (indirect leaching) resulting in sulfuric acid and ferric ions, where the acid and oxidant media solubilizes the metals. Also, the microorganism can react directly into the material, releasing the metals to the solution (direct leaching). For example, Mishra et al. (2008) evaluated the bioleaching from LCO type cathode by *Acidithiobacillus ferrooxidans*, where Co and Li extraction achieved 55% and 10%, respectively, after 20 days of reaction at pH 2.5, solid-liquid ratio equals to 1/10 and 3 g/L of ferrous iron [167].

Niu et al. (2014) studied the effect of pulp density and increased the leaching of Co and Li to 72% and 89% at 1/50 (solid-liquid ratio) [168]. Results of Ni extraction reported by Ijadi Bajestani et al. (2014) in bioleaching of NiCd and NiMH batteries indicated that the *Acidithiobacillus ferrooxidans* might be used for Ni extraction [169].

Xin et al. (2016) studied the bioleaching of LFP, LMO, and NMC types of cathode using *Acidithiobacillus thiooxidans* and *Leptospirillum ferriphilum* in solid-liquid ratio equals to 1/100. Li, Mn, Co, and Ni extraction achieved 99% after 9 days [170]. In contrast, direct acid leaching by $H_2SO_4$ takes at least 90 min to achieve similar results [42]. Roy et al. (2021) found the same Co bioleaching results of Xin et al. (2016) for LCO type cathode recycling in a solid-liquid ratio equal to 1/100 [171].

A few authors suggest the use of microorganic to produce organic acids to promote further the leaching of Li-ion batteries, such as citric acid, malic acid, aspartic acid, and succinic acid. Leaching efficiencies for Ni, Co, Mn, and Li can achieve over 95% [172–175]. For example, Bahaloo-Horeh et al. (2018) evaluated the adapted *Aspergillus niger* for leaching metals from mobile phone batteries, where organic acids are generated reacting with the cathode material—gluconic, oxalic, malic, and citric acids [176].

Also, Dolker & Pant (2019) studied the *Lysinibacillus* sp. mixed with citric acid as a hybrid system for Co recovery. First, the cathode material was released from Al foils using citric acid only. Then, a mixed reaction was used, where Co was first leached and then adsorbed by *Lysinibacillus* sp. biomass. The process efficiency was 98% [177]. For this reason, biohydrometallurgy can be considered an ideal route for industrial application, but it is necessary to study for selective separation of Co after bioleaching. Besides, slower kinetics is an issue when compared to the hydrometallurgy process. However, environmental benefits and low costs may benefit to scale-up from lab to industrial scale [15].

Finally, Stopic and Friedrich [178] reported a comparative overview of Co-recovery from primary and secondary materials including the hydrometallurgical treatment of Li-ion batteries. Biohydrometallurgy for separation steps must be improved to obtain high-pure products.

## 4. Conclusions

The present study aimed at the literature review of Co recovery from Li-ion batteries towards sustainable development. The mature hydrometallurgy techniques were explored, such as acid leaching by inorganic and organic acids, solvent extraction, ion-exchange resins, and precipitation. Despite what's well-known about inorganic acid in leaching, the use of organic acids have achieved similar results, and efforts should focus on the organic matrix purification step. Moreover, organic compouns would be more sustainable than inorganic acids, less dangerous and cheaper. In separation step, solvent extraction achieves higher separation degrees, which is important to obtain high pure products. On the other hand, improvements in ion exchange resins process may achieve similar results. Precipitation technique is indicated, as reported by the literature, to obtain a new cathode after acid leaching. The emerging technologies ionic liquids, deep eutectic solvent, supercritical fluids, nanotechnology, and biohydrometallurgy are presented. There is a lack in the literature about such techniques for Co extraction from spent Li-ion batteries. Future studies must be carried out to analyse the pros and cons of each technique. The discussion throughout the manuscript has demonstrated that these technologies have potential applications for Li-ion batteries. This literature review addresses the development of new recycling processes towards sustainable development, which will be the main target of the next studies.

**Author Contributions:** Conceptualization, methodology, formal analysis, investigation, data curation, writing—original draft preparation, A.B.B.J., D.C.R.E. and J.A.S.T.; writing—review and editing, S.S. and B.F. All authors have read and agreed to the published version of the manuscript.

**Funding:** This research received no external funding.

**Institutional Review Board Statement:** Not applicable.

**Informed Consent Statement:** Not applicable.

**Data Availability Statement:** Not applicable.

**Acknowledgments:** To the University of Sao Paulo to support this project. To Fundação de Amparo à Pesquisa do Estado de São Paulo and Capes (2019/11866-5, 2012/51871-9, 2018/03483-6, 2018/11417-3) for the financial support in previous research. We thank sincerely the Editorial board members and three anonymous reviewers for their constructive comments.

**Conflicts of Interest:** The authors declare no conflict of interest.

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
