# Peer review of "Cobalt Recovery from Li-Ion Battery Recycling: A Critical Review"

_metals, doi:10.3390/met11121999_

Round 1

Reviewer 1 Report

There is no mention of the change of purity of cobalt after various hydrometallurgical processes.

State and add the above mentions on the paper.

Author Response

Dear reviewer and editor Dr. Aaron Han,

Many thanks for your valuable comments. We learned greatly from your rigorous attitude towards scientific research. Your comments have an important guide for improving our article. We have studied your comments carefully and have modified the whole manuscript according to your comments. Just to be clear, the manuscript was entirely revised and we believe that all the explanations are now to the point and brief. In our point of view, the revised manuscript is now more acceptable than previously.

Nevertheless, please do not hesitate to send your suggestions and comments to us and we are very glad to receive these because we can always learn a lot from your valuable suggestions and comments. All changes are highlighted in green in the revised manuscript.

Regarding your Comments and Suggestions for Authors:

There is no mention of the change of purity of cobalt after various hydrometallurgical processes.

State and add the above mentions on the paper.

Our answer: Thank you for your valuable comment. We added in the manuscript the information available about purity of cobalt solution. Not all articles reported the degree of purity of the products. In the case of leaching/extraction step, the purpose is the solubilization of all target metals, and further selective separation.

Reviewer 2 Report

The paper is a week structured and clear review on cobalt recovery, which I like to read. The paper can be accepted in the proposed form. However, some lacks of references in the text (equation number, citations) were not transferred to the final file and appears as missing fields in the text.

Author Response

Dear reviewer and our editor Dr. Aaron Han,

Many thanks for your valuable comments. We learned greatly from your rigorous attitude towards scientific research. Your comments have an important guide for improving our article. We have studied your comments carefully and have modified the whole manuscript according to your comments. Just to be clear, the manuscript was entirely revised and we believe that all the explanations are now to the point and brief. In our point of view, the revised manuscript is now more acceptable than previously.

Nevertheless, please do not hesitate to send your suggestions and comments to us and we are very glad to receive these because we can always learn a lot from your valuable suggestions and comments. All changes are highlighted in green in the revised manuscript.

Comments and Suggestions for Authors

The paper is a week structured and clear review on cobalt recovery, which I like to read. The paper can be accepted in the proposed form. However, some lacks of references in the text (equation number, citations) were not transferred to the final file and appears as missing fields in the text.

Answer: Thank you for your valuable comments. We revised the manuscript as suggested.

Reviewer 3 Report

Overall, the paper is poorly written. The structure of sentences and wording used makes it is very difficult to read. In some cases, it’s a simple matter of removing an unnecessary word such as “the” or “for” or replacing a certain word with one that is more suitable for the flow of the sentence. For instance, ‘Co obtaining’ (a term used throughout the paper) can be replaced with “Co recovery”. However, in a majority of the cases, they need to be rewritten or rearranged as the sentences do not make sense for example page 2 line 46 and 47 ‘The control of a single country's most significant production and resources justifies the concerns about Co supply worldwide’ and page 2 line 65 and 66 ‘Based on energy generation by electrochemical process, these batteries accumulate Li ions in the anode material flow in direction to the cathode in discharging procedure’. Furthermore, in some instances the language used is not of a professional manner for example page 2 line 93 the phrase ‘super cost’. For a reader with prior knowledge in the topic it is possible to garner an understanding of what the authors are trying to convey, however, it has not been articulated well and the paper cannot be considered a professional piece of academic writing.

After approximately page 7 there is noticeable improvement in the writing, however, there are still numerous langue and structural errors that need to be corrected.

While the authors have consulted a large breadth of previous works and have summarised these, in order to publish the paper, it needs to be rewritten in a clear and concise manner so that it is coherent. It is also recommended that the authors proof the paper carefully to ensure figures given are correct, for example the value ‘3711%’ on page 3 line 103 does not seem correct.

Author Response

Dear reviewer and our editor Dr. Aaron Han,

Many thanks for your valuable comments. We learned greatly from your rigorous attitude towards scientific research. Your comments have an important guide for improving our article. We have studied your comments carefully and have modified the whole manuscript according to your comments. Just to be clear, the manuscript was entirely revised and we believe that all the explanations are now to the point and brief. In our point of view, the revised manuscript is now more acceptable than previously.

Nevertheless, please do not hesitate to send your suggestions and comments to us and we are very glad to receive these because we can always learn a lot from your valuable suggestions and comments. All changes are highlighted in green in the revised manuscript.

Overall, the paper is poorly written. The structure of sentences and wording used makes it is very difficult to read. In some cases, it’s a simple matter of removing an unnecessary word such as “the” or “for” or replacing a certain word with one that is more suitable for the flow of the sentence. For instance, ‘Co obtaining’ (a term used throughout the paper) can be replaced with “Co recovery”. However, in a majority of the cases, they need to be rewritten or rearranged as the sentences do not make sense for example page 2 line 46 and 47 ‘The control of a single country's most significant production and resources justifies the concerns about Co supply worldwide’ and page 2 line 65 and 66 ‘Based on energy generation by electrochemical process, these batteries accumulate Li ions in the anode material flow in direction to the cathode in discharging procedure’. Furthermore, in some instances the language used is not of a professional manner for example page 2 line 93 the phrase ‘super cost’. For a reader with prior knowledge in the topic it is possible to garner an understanding of what the authors are trying to convey, however, it has not been articulated well and the paper cannot be considered a professional piece of academic writing.

Answer: Thank you for the comments. We revised the manuscript and all your considerations were taken into account.

After approximately page 7 there is noticeable improvement in the writing, however, there are still numerous langue and structural errors that need to be corrected.

Answer: Thank you for the comments. We made the corrections as highlighted. The manuscript was revised before new submission.

While the authors have consulted a large breadth of previous works and have summarised these, in order to publish the paper, it needs to be rewritten in a clear and concise manner so that it is coherent. It is also recommended that the authors proof the paper carefully to ensure figures given are correct, for example the value ‘3711%’ on page 3 line 103 does not seem correct.

Answer: Thank you for the comments. We revised the manuscript and all changes are highlighted. About the ´3711%´ value, it is the real increase rate (see doi:10.1016/j.jenvman.2021.113091)

Reviewer 4 Report

This manuscript reviews the techniques for Co recovering from spent Li-ion battery, for instance,acid leaching (inorganic and organic), separation (solvent extraction, ion exchange resins, and precipitation), and emerging technologies. The relevance of the topic is beyond doubt. However, the current version of this manuscript is unable to be accepted for publication.

  1. The whole article does not review on the recovery of Co from waste Li-ion batteries by pyrometallurgy or pyro-hydrometallurgy combined process. as we all known, it is an important process to recovery metal from spent materials by pyrometallurgy.

  1. The introduction is too long. Do you want to add some contents to a new chapter to introduce the battery and its pretreatments? Such as dischargement, crushing and others.

3 In some references, the reasons for the low leaching efficiency of cobalt need to explain, such as [47], [53] and [63].

  1. The Co of reference [47] is leached from NiO·Mn2O3·Co3O4 rather than the original NMC battery. This reference is not suitable as an example of sulfuric acid leaching, and may be an example of the disadvantages of pyrometallurgical treatment of spent batteries.

  1. The conclusion of this manuscript hasn't been presented deeply.

Author Response

Dear reviewer and our editor Dr. Aaron Han,

Many thanks for your valuable comments. We learned greatly from your rigorous attitude towards scientific research. Your comments have an important guide for improving our article. We have studied your comments carefully and have modified the whole manuscript according to your comments. Just to be clear, the manuscript was entirely revised and we believe that all the explanations are now to the point and brief. In our point of view, the revised manuscript is now more acceptable than previously.

Nevertheless, please do not hesitate to send your suggestions and comments to us and we are very glad to receive these because we can always learn a lot from your valuable suggestions and comments. All changes are highlighted in green in the revised manuscript.

Comments:

This manuscript reviews the techniques for Co recovering from spent Li-ion battery, for instance,acid leaching (inorganic and organic), separation (solvent extraction, ion exchange resins, and precipitation), and emerging technologies. The relevance of the topic is beyond doubt. However, the current version of this manuscript is unable to be accepted for publication.

  1. The whole article does not review on the recovery of Co from waste Li-ion batteries by pyrometallurgy or pyro-hydrometallurgy combined process. as we all known, it is an important process to recovery metal from spent materials by pyrometallurgy.

Answer: Thank you for the comments. The manuscript focused on hydrometallurgical processes due to the low greenhouse gas emission, less energy consumption and obtaining of high-pure products (as stated in page 4 line 138-149)

2.The introduction is too long. Do you want to add some contents to a new chapter to introduce the battery and its pretreatments? Such as dischargement, crushing and others.

Answer: In the current format, there is no need to add contents about pretreatments. We already discussed in deapth in our last publication about Li-ion battery (see https://doi.org/10.1016/j.jenvman.2021.113091)

  1. In some references, the reasons for the low leaching efficiency of cobalt need to explain, such as [47], [53] and [63].

Answer: The losses of Co in reference [47] is related to the Li dissolution.

The goal of reference [53] was the Li extraction.

In the case of reference [63], acetic acid has low capacity of Co extraction.

  1. The Co of reference [47] is leached from NiO·Mn2O3·Co3O4 rather than the original NMC battery. This reference is not suitable as an example of sulfuric acid leaching, and may be an example of the disadvantages of pyrometallurgical treatment of spent batteries.

Answer: Thank you for the comment. We added the observation in the manuscript, as highlighted in green.

  1. The conclusion of this manuscript hasn't been presented deeply.

Answer: We revised the Conclusion section in detail.

Reviewer 5 Report

The authors have attempted a comprehensive review on the recovery of cobalt from waste Li-ion batteries. It covers leaching, separation, and also modern technologies with nearly 180 papers. The most important aspect of a technical review paper is its sensible and relevant commentary and is not merely a summary. This paper presents a laundry list of papers without critical review which should include a relevant description of topic, overall perspective, argument, purpose, and future direction. It does contain critical reviews but most of them are criticisms given by other authors than their own.

This review recommends that the conclusion should be expanded to cover the authors’ critical view on various topic matters. For example, is the ordinary acid leaching better than organic acid leaching, why and why not. What is between solvent extraction, ion-exchange, and precipitation? What is lack in theoretical aspects in this area? What should be the future direction in the recovery of Co from Li-batteries and so on.

Author Response

Dear reviewers and our editor Dr. Aaron Han,

Many thanks for your valuable comments. We learned greatly from your rigorous attitude towards scientific research. Your comments have an important guide for improving our article. We have studied your comments carefully and have modified the whole manuscript according to your comments. Just to be clear, the manuscript was entirely revised and we believe that all the explanations are now to the point and brief. In our point of view, the revised manuscript is now more acceptable than previously.

Nevertheless, please do not hesitate to send your suggestions and comments to us and we are very glad to receive these because we can always learn a lot from your valuable suggestions and comments. All changes are highlighted in green in the revised manuscript.

Comments:

The authors have attempted a comprehensive review on the recovery of cobalt from waste Li-ion batteries. It covers leaching, separation, and also modern technologies with nearly 180 papers. The most important aspect of a technical review paper is its sensible and relevant commentary and is not merely a summary. This paper presents a laundry list of papers without critical review which should include a relevant description of topic, overall perspective, argument, purpose, and future direction. It does contain critical reviews but most of them are criticisms given by other authors than their own.

Answer: Thank you for the comments. All discussions presented throughout the manuscript were done by the authors after the extense literature review. We added more discussion in the manuscript as highlighted in green.

This review recommends that the conclusion should be expanded to cover the authors’ critical view on various topic matters. For example, is the ordinary acid leaching better than organic acid leaching, why and why not. What is between solvent extraction, ion-exchange, and precipitation? What is lack in theoretical aspects in this area? What should be the future direction in the recovery of Co from Li-batteries and so on.

Answer: Thank you for the comments. We made improvements in the Conclusion section as recommended.

Round 2

Reviewer 3 Report

The quality of the paper has improved and is clearly evident. However, it recommend that the authors proof the paper as there are some minor errors. For example:

  1. Page 10 line 285 and 286 “For instance, in nickel laterite processing, for instance, the bis(2,4,4-trimethylpentyl)phosphonic acid extractants are used in sulfate…”, the second ‘for instance’ needs to be removed.
  2. Page 10 line 306 and 307 “Tsakiridis & Tsakiridis & Agatzini-Leonardou (2005) studied…”, ‘Tsakiridis’ is repeated twice.
  3. Page 14 line 410 “…obtain CoC2O4 as solid leaching Li ions in solution”, shouldn’t the word ‘leaching’ be ‘leave’?
  4. Page 14 line 438 and 439 “…which followed the heating at 450ËšC and calcination at 950ËšC to obtain the cathode material…”, it should be rephrased to ‘…which is followed by heating at…’.

Author Response

Dear Reviewer,

thank you very much for your invested time and valuable comments. According to your comments we are sending our answers:

The quality of the paper has improved and is clearly evident. However, it recommend that the authors proof the paper as there are some minor errors. For example:

Page 10 line 285 and 286 “For instance, in nickel laterite processing, for instance, the bis(2,4,4-trimethylpentyl)phosphonic acid extractants are used in sulfate…”, the second ‘for instance’ needs to be removed.

Answers: Thank you for the comment. We made the correction.

Page 10 line 306 and 307 “Tsakiridis & Tsakiridis & Agatzini-Leonardou (2005) studied…”, ‘Tsakiridis’ is repeated twice.

Answers: Thank you for the comment. We made the correction.

Page 14 line 410 “…obtain CoC2O4 as solid leaching Li ions in solution”, shouldn’t the word ‘leaching’ be ‘leave’?

Answers: Thank you for the comment. We changed the word.

Page 14 line 438 and 439 “…which followed the heating at 450ËšC and calcination at 950ËšC to obtain the cathode material…”, it should be rephrased to ‘…which is followed by heating at…’.

Answers: Thank you for the recommendation. We rephrased the sentence as suggested.

Reviewer 4 Report

The quality of the paper has improved clearly, but some spell mistakes should be  corrected.

Author Response

Dear Reviewer,

thank you for your invested time and valuable comments. According to your comments I am sending our answers:

The quality of the paper has improved clearly, but some spell mistakes should be  corrected.

Answer: We have corrected some spell mistakes in text. They are important to improve the quality of our manuscript.

Reviewer 5 Report

The authors have made some relevant and useful response to the reviewer’s comments and suggestions. The paper does contain useful information. This reviewer recommends that the paper be accepted for publication. It does need to correct some typos such as “futures studies….”

Author Response

Dear Reviewer, thank you very much for your invested time and valuable comments. According to your comments we are sending our answer.

The authors have made some relevant and useful response to the reviewer’s comments and suggestions. The paper does contain useful information. This reviewer recommends that the paper be accepted for publication. It does need to correct some typos such as “futures studies….”

Answer: The comments really helped to improve the quality of the manuscript. We have corrected "futures studies".